# Improving SAT Solver Heuristics with Graph Networks and Reinforcement Learning

## Abstract

We present GQSAT, a branching heuristic in a Boolean SAT solver trained with value-based reinforcement learning (RL) using Graph Neural Networks for function approximation. Solvers using GQSAT are complete SAT solvers that either provide a satisfying assignment or a proof of unsatisfiability, which is required for many SAT applications. The branching heuristic commonly used in SAT solvers today suffers from bad decisions during their warm-up period, whereas GQSAT has been trained to examine the structure of the particular problem instance to make better decisions at the beginning of the search. Training GQSAT is data efficient and does not require elaborate dataset preparation or feature engineering to train. We train GQSAT on small SAT problems using RL interfacing with an existing SAT solver. We show that GQSAT is able to reduce the number of iterations required to solve SAT problems by 2-3X, and it generalizes to unsatisfiable SAT instances, as well as to problems with 5X more variables than it was trained on. We also show that, to a lesser extent, it generalizes to SAT problems from different domains by evaluating it on graph coloring. Our experiments show that augmenting SAT solvers with agents trained with RL and graph neural networks can improve performance on the SAT search problem.

## 1 Introduction

Boolean satisfiability (SAT) is an important problem for both industry and academia impacting various fields, including circuit design, computer security, artificial intelligence, automatic theorem proving, and combinatorial optimization. As a result, modern SAT solvers are well-crafted, sophisticated, reliable pieces of software that can scale to problems with hundreds of thousands of variables (Ohrimenko et al., 2009).

SAT is known to be NP-complete (Karp, 1972), and most state-of-the-art open-source and commercial solvers rely on multiple *heuristics* to speed up the exhaustive search, which is otherwise intractable. These heuristics are usually meticulously crafted using expert domain knowledge and are often iterated on using trial and error. In this paper, we investigate how we can use machine learning to improve upon an existing branching heuristic without leveraging domain expertise.

We present Graph-Q-SAT (GQSAT), a branching heuristic in a Conflict Driven Clause Learning (Marques-Silva & Sakallah, 1999; Bayardo Jr & Schrag, 1997, CDCL) SAT solver trained with value-based reinforcement learning (RL), based on DQN (Mnih et al., 2015). GQSAT uses a graph representation of SAT problems similar to Selsam et al. (2018) which provides permutation and variable relabeling invariance. It uses a Graph Neural Network (Gori et al., 2005; Battaglia et al., 2018, GNN) as a function approximator to provide generalization as well as support for a dynamic state-action space. GQSAT uses a simple state representation and a binary reward that requires no feature engineering or problem domain knowledge. GQSAT modifies only part of the CDCL based solver, keeping it *complete*, i.e. always leading to a correct solution.

We demonstrate that GQSAT outperforms Variable State Independent Decaying Sum (Moskewicz et al., 2001, VSIDS), most frequently used CDCL branching heuristic, reducing the number of iterations required to solve SAT problems by 2-3X. GQSAT is trained to examine the structure of the particular problem instance to make better decisions at the beginning of the search whereas the VSIDS heuristic suffers from bad decision during the warm-up period. We show that our method generalizes to problems five times larger than those it was trained on. We also show that our method

generalizes across problem types from SAT to unSAT. We also show, to a lesser extent, it generalizes to SAT problems from different domains, such as graph colouring. Finally, we show that some of these improvements are achieved even when training is limited to single SAT problem demonstrating data efficiency of our method. We believe GQSAT is a stepping stone to a new generation of SAT solvers leveraging data to build better heuristics learned from past experience.

## 2 BACKGROUND

### 2.1 SAT PROBLEM

A SAT problem involves finding variable assignments such that a propositional logic formula is satisfied or showing that such an assignment does not exist. A propositional formula is a Boolean expression, including Boolen variables, ANDs, ORs and negations. 'x' or 'NOT x' make up a literal. It is convenient to represent Boolean formulas in conjunctive normal form (CNF), i.e., conjunctions (AND) of clauses, where a clause is a disjunction (OR) of literals. An example of a CNF is $(x_1 \vee \neg x_2) \wedge (x_2 \vee \neg x_3)$, where $\wedge, \vee, \neg$ are AND, OR, and negation respectively. This CNF has two clauses: $(x_1 \vee \neg x_2)$ and $(x_2 \vee \neg x_3)$. In this work, we use SAT to both denote the Boolean Satisfiability problem and a satisfiable instance, which should be clear from the context. We use unSAT to denote unsatisfiable instances.

There are many types of SAT solvers. In this work, we focus on CDCL solvers, MiniSat (Sorensson & Een, 2005) in particular, because it is an open source, minimal, but powerful implementation. A CDCL solver repeats the following steps: every iteration it picks a literal, assigns a variable a binary value. This is called a decision. After deciding, the solver simplifies the formula building an implication graph, and checks whether a conflict emerged. Given a conflict, it can infer (learn) new clauses and backtrack to the variable assignments where the newly learned clause becomes unit (consisting of a single literal) forcing a variable assignment which avoids the previous conflict. Sometimes, CDCL solver undoes all the variable assignments keeping the learned clauses to escape futile regions of the search space. This is called a restart.

We focus on the branching heuristic because it is one of the most heavily used during the solution procedure. The branching heuristic is responsible for picking the variable and assigning some value to it. VSIDS (Moskewicz et al., 2001) is one of the most used CDCL branching heuristics. It is a counter-based heuristic which keeps a scalar value for each literal or variable (MiniSat uses the latter). These values are increased every time a variable gets involved in a conflict. The algorithm behaves greedily with respect to these values called *activities*. Activities are usually initialized with zeroes (Liang et al., 2015).

### 2.2 REINFORCEMENT LEARNING

We formulate the RL problem as a Markov decision process (MDP). An MDP is a tuple $\langle \mathcal{S}, \mathcal{A}, \mathcal{R}, \mathcal{T}, \rho, \gamma \rangle$ with a set of states $\mathcal{S}$, a set of actions $\mathcal{A}$, a reward function $\mathcal{R} = R(s, a, s')$ and the transition function $\mathcal{T} = p(s, a, s')$, where $p(s, a, s')$ is a probability distribution, $s, s' \in \mathcal{S}$, $a \in \mathcal{A}$. $\rho$ is the probability distribution over initial states. $\gamma \in [0, 1)$ is the discount factor responsible for trading off the preferences between the current immediate reward and the future reward. In the case of *episodic tasks*, the state space is split into the set of non-terminal states and the terminal state $\mathcal{S}^+$. To solve an MDP means to find an optimal policy, a mapping which outputs an action or distribution over actions given the state, such that we maximize the expected discounted cumulative return $R = \mathbb{E}[\sum_{t=0}^{\infty} \gamma^t r_t]$, where $r_t = R(s_t, a_t, s_{t+1})$ is the reward for the transition from $s_t$ to $s_{t+1}$.

In Section 3 we apply deep $Q$-networks (Mnih et al., 2015, DQN), a value-based RL algorithm that approximates an optimal $Q$-function, an action-value function that estimates the sum of future rewards after taking an action $a$ in state $s$ and following the optimal policy $\pi$ thereafter: $Q^*(s, a) = \mathbb{E}_{\pi, \mathcal{T}, \rho}[R(s, a, s') + \gamma \max_{a'} Q^*(s', a')]$. A mean squared temporal difference (TD) error is used to make an update step: $L(\theta) = (Q_\theta(s, a) - r - \gamma \max_{a'} Q_{\bar{\theta}}(s', a'))^2$. $Q_{\bar{\theta}}$ is called a target network (Mnih et al., 2015). It is used to stabilise DQN by splitting the decision and evaluation operations. Its weights are copied from the main network $Q_\theta$ after each $k$ minibatch updates.

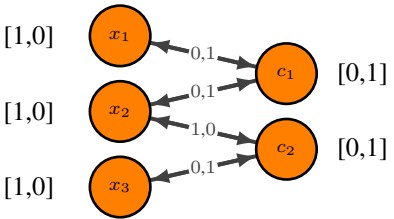 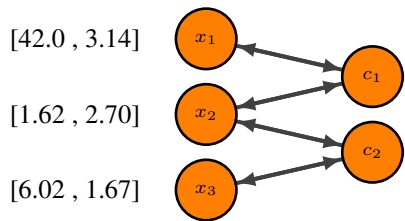

(a) Bipartite graph representation of the Boolean formula $(x_1 \vee x_2) \wedge (\neg x_2 \vee x_3)$. The numbers next to the vertices distinguish variables and clauses. Edge labels encode literal polarities. The global component does not contain any information about the state.

(b) Graph Q-function values for setting variables to *true* and *false* respectively. The action is chosen using $\arg\max$ across all Q values of variable nodes. Since GNN work on directed graphs, we add two directional edges to connect two nodes.

Figure 1: State-action space representation of GQSAT

## 2.3 GRAPH NEURAL NETWORKS

We use Graph Neural Networks (Gori et al., 2005, GNN) to approximate our $Q$-function due to their input size, structure, and permutation invariance. We use the formalism of Battaglia et al. (2018) which unifies most existing GNN approaches. Under this formalism, GNN is a set of functions that take a labeled graph as input and output a graph with modified labels but the same topology.

Here, a graph is a directed graph $\langle V, E, U \rangle$, where $V$ is the set of vertices, $E$ is the set of edges with $e = (s, r) \in E$, $s, r \in V$, and $U$ is a global attribute. The global attribute contains information, relevant to the whole graph. We call vertices, edges and the global attribute entities. Each entity has its features vectors. A GNN changes this features as a result of its operations.

A graph network can be seen as a set of six functions: three update functions and three aggregation functions. The information propagates between vertices along graph edges. Update functions compute new entity labels. Aggregation functions exist to ensure the GNN's ability to process graphs of arbitrary topologies, compressing multiple entities features into vectors of fixed size. GNN blocks can be combined such that the output of one becomes input of the other. For example, the Encode-Process-Decode architecture (Battaglia et al., 2018) processes the graph in a recurrent way, enabling information propagation between remote vertices.

## 3 GQSAT

As noted in Section 2.2, we use the MDP formalism for our purposes. Each state of our MDP consists of unassigned variables and unsatisfied clauses containing these variables. The initial state distribution is a distribution over all possible SAT problems. Our problem has an episodic nature with a clear terminal state: when a satisfying assignment is found or the algorithm has exhausted all the possible options proving unSAT. The action set includes two actions for each unassigned variable: assigning it to true or false. We modify the MiniSat-based environment of Wang & Rompf (2018) which is responsible for the transition function. It takes the actions, modifies its implication graph internally and returns a new state, containing newly learned clauses and without the variables removed after the propagation. Strictly speaking, this state is not fully observable. In the case of a conflict, the solver undoes the assignments for variables that are not in the agent's observation. However, in practice, this should not inhibit the goal of quickly pruning the search tree: the information in the state is enough to pick a variable that leads to more propagations in the remaining formula.

We use a simple reward function: the agent gets a negative reinforcement $p$ for each non-terminal transition and 0 for reaching the terminal state. This reward encourages an agent to finish an episode as quickly as possible and does not require elaborate reward shaping to start using GQSAT.

## 3.1 STATE REPRESENTATION

We represent a SAT problem as a graph similar to Selsam et al. (2018). We make it more compact, using vertices to denote variables instead of literals. We use nodes to encode clauses as well.

Our state representation is simple and does not require scrupulous feature engineering. An edge $(x_i, c_i)$ means that a clause $c_i$ contains literal $x_i$. If a literal contains a negation, a corresponding edge has a $[1, 0]$ label and $[0, 1]$ otherwise. GNN process directed graphs so we create two directed edges with the same labels: from a variable to a clause and vice-versa. Vertex features are two dimensional one-hot vectors, denoting either a variable or a clause. We do not provide any other information to the model. The global attribute input is empty and is only used for message passing. Figure 1a gives an example of the state for $(x_1 \lor x_2) \land (\neg x_2 \lor x_3)$.

## 3.2 Q-FUNCTION REPRESENTATION

We use the encode-process-decode architecture (Battaglia et al., 2018), which we discuss in more detail in Appendix B.1. Similarly to Bapst et al. (2019), our GNN labels variable vertices with $Q$-values. Each variable vertex has two actions: pick the variable and set it to true or false as shown on Figure 1b. We choose the action which gives the maximum Q-value across all variable vertices. The graph contains only unassigned variables so all actions are valid. We use common DQN techniques such as memory replay, target network and $\epsilon$-greedy exploration. To expose the agent to more episodes and prevent it from getting stuck, we cap the maximum number of actions per episode. This is similar to the *episode length* parameter in *gym* (Brockman et al., 2016).

## 3.3 TRAINING AND EVALUATION PROTOCOL

We train our agent using Random 3-SAT instances from the SATLIB benchmark (Hoos & Stützle, 2000). To measure generalization, we split these data into train, validation and test sets. The train set includes 800 problems, while the validation and test sets are 100 problems each. We provide more details about dataset in Appendinx B.2.

To illustrate the problem complexities, Table 1 provides the number of steps it takes MiniSat to solve the problem. Each random 3-SAT problem is denoted as SAT-X-Y or unSAT-X-Y, where SAT means that all problems are satisfiable, unSAT stands for all problems are unsatisfiable. X and Y stands for the number of variables and clauses in the initial formula.

While random 3-SAT problems have relatively small number of variables/clauses, they have an interesting property which makes them more challenging for a solver. For this dataset, the ratio of clauses to variables is close to 4.3:1 which is near the *phase transition*, when it is hard to say whether the problem is SAT or unSAT (Cheeseman et al., 1991). In 3-SAT problems each clause has exactly 3 variables, however, learned clauses might be of arbitrary size and GQSAT is able to deal with it.

We use Median Relative Iteration Reduction (MRIR) w.r.t. MiniSat as our main performance metric which is a number of iterations it takes MiniSat to solve a problem divided by GQSAT's number of iterations. Similarly to the *median human normalised score* adopted in the Atari domain, we use the median instead of the mean to avoid the situation when the outliers skew the mean considerably. By one iteration we mean one *decision*, i.e. choosing a variable and setting it to a value. MRIR is the median across all the problems in the dataset. We compare ourselves with the best MiniSat results having run MiniSat with and without restarts. We cap the number of decisions our method takes at the beginning of the solution procedure and then we give control to MiniSat.

When training, we evaluate the model every 1000 batch updates on the validation subsets of the same distribution as the train dataset and pick the one with the best validation results. After that, we evaluate this model on the test dataset and report the results. For each of the model we do 5 training runs and report the average MRIR results, the maximum and the minimum.

We implement our models using Pytorch (Paszke et al., 2017) and Pytorch Geometric (Fey & Lenssen, 2019). We provide all the hyperparameters needed to reproduce our results in Appendix B. We will release our experimental code as well as the MiniSat *gym* environment.

Table 1: Number of iterations (no restarts) it takes MiniSat to solve random 3-SAT instances.

| dataset | median | mean |
|---|---|---|
| SAT 50-218 | 38 | 42 |
| SAT 100-430 | 232 | 286 |
| SAT 250-1065 | 62 192 | 76 120 |
| unSAT 50-128 | 68 | 68 |
| unSAT 100-430 | 587 | 596 |
| unSAT 250-1065 | 178 956 | 182 799 |

Table 2: MRIR for GQSAT trained on SAT-50-218. Evaluation for SAT-50-218 is on a separate test data not seen during training.

| dataset | mean | min | max |
|---|---|---|---|
| SAT 50-218 | 2.46 | 2.26 | 2.72 |
| SAT 100-430 | 3.94 | 3.53 | 4.41 |
| SAT 250-1065 | 3.91 | 2.88 | 5.22 |
| unSAT 50-128 | 2.34 | 2.07 | 2.51 |
| unSAT 100-430 | 2.24 | 1.85 | 2.66 |
| unSAT 250-1065 | 1.54 | 1.30 | 1.64 |

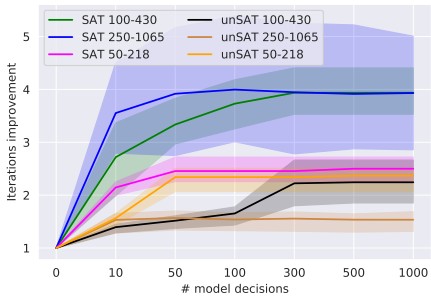

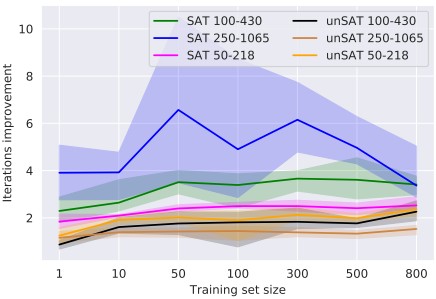

Figure 2: GQSAT number of maximum first decisions vs performance. GQSAT shows improvement starting from 10 iterations confirming our hypothesis of VSIDS initialization problem. Best viewed in colour.

Figure 3: Dataset size effect on generalization. While GQSAT profits from having more data in most of the cases, it is able to generalize even from one data point. Model is trained on SAT-50-218. Best viewed in colour.

## 4 EXPERIMENTAL RESULTS

### 4.1 IMPROVING UPON VSIDS

In our first experiment, we consider whether it is possible improve upon VSIDS using no domain knowledge, a simple state representation, and a simple reward function. The first row in Table 2 gives us a positive answer to that question. DQN equipped with GNN as a function approximation solves the problems in fewer than half the iterations of MiniSat.

GQSAT makes decisions resulting in more propagations, i.e., inferring variable values based on other variable assignments and clauses. This helps GQSAT prune the search tree faster. For SAT-50-218, GQSAT does on average 2.44 more propagations than MiniSat (6.62 versus 4.18). We plot the average number of variable assignments for each problem individually in the Appendix A.

These results raise the question: Why does GQSAT outperform VSIDS? VSIDS is a counter-based heuristic that takes time to warm up. Our model, on the other hand, perceives the whole problem structure and can make more informed decisions from step one. To check this hypothesis, we vary the number of decisions our model makes at the beginning of the solution procedure before we hand the control back to VSIDS. The results of the experiment in Figure 2 support this hypothesis. Even if our model is used for only the first ten iterations, it still improves performance over VSIDS.

One strength of GQSAT is that VSIDS keeps being updated while the decisions are made with GQSAT. We believe that GQSAT complements VSIDS by providing better quality decisions in the initial phase while VSIDS is warming up. Capping the number of model calls can significantly reduce the main bottleneck of our approach – wall clock time spent on model evaluation. Optimizing for speed was not our focus, however even with the current unoptimized implementation, if we use the model for the first 500 iterations and assuming this gives us a 2x reduction in total iterations, our approach is competitive if it takes more than 20 seconds for a base solver to solve the problem.

Table 3: SAT-50 models performance on SATLIB flat graph coloring benchmark. Median Relative Iteration Reduction (MRIR) is w.r.t. MiniSat with restarts, since MiniSat performs better in this mode for this benchmark.

| dataset | variables | clauses | MiniSat median iterations | GQSAT MRIR | | |
| --- | --- | --- | --- | --- | --- | --- |
| | | | | average | min | max |
| flat-30-60 | 90 | 300 | 10 | 1.51 | 1.25 | 1.65 |
| flat-50-115 | 150 | 545 | 15 | 1.36 | 0.47 | 1.80 |
| flat-75-80 | 225 | 840 | 29 | 1.40 | 0.31 | 2.06 |
| flat-100-239 | 300 | 1117 | 55 | 1.44 | 0.31 | 2.38 |
| flat-125-301 | 375 | 1403 | 106 | 1.02 | 0.32 | 1.87 |
| flat-150-360 | 450 | 1680 | 179 | 0.76 | 0.37 | 1.40 |
| flat-175-417 | 525 | 1951 | 272 | 0.67 | 0.44 | 1.36 |
| flat-200-479 | 600 | 2237 | 501 | 0.67 | 0.54 | 0.87 |

## 4.2 GENERALIZATION PROPERTIES OF GQSAT

### 4.2.1 GENERALIZATION ACROSS PROBLEM SIZES

Table 2 shows that GQSAT has no difficulties generalizing to bigger problems, showing almost 4x improvement in iterations for the dataset 5 times bigger than the training set. GQSAT on average leads to more variable assignments changes per step, e.g., 7.58 vs 5.89 on SAT-100-430. It might seem surprising that the model performs better for larger problems. However, our performance metric is relative. An increase in score for different problem sizes might also mean that the base solver scales worse than our method does for this benchmark.

### 4.2.2 GENERALIZATION FROM SAT TO UNSAT

An important characteristic of GQSAT is that the problem formulation and representation makes it possible to solve unSAT problems when training only on SAT, which was problematic for some of the existing approaches (Selsam et al., 2018).

The performance is, however, worse than the performance on satisfiable problems. On the one hand, SAT and unSAT problems are different. When the solver finds one satisfying assignment, the problem is solved. For unSAT, the algorithm needs to exhaust all possible options to prove that there is no such assignment. On the other hand, there is one important similarity between these two types of problems – an algorithm has to prune the search tree as fast as possible. Our measurements of average propagations per step demonstrate that GQSAT learns how to prune the tree more efficiently than VSIDS (6.36 vs 4.17 for unSAT-50-218).

### 4.2.3 GENERALIZATION ACROSS PROBLEM STRUCTURES

SAT problems have distinct structures. The graph representation of a random 3-SAT problem looks much different than that of a graph coloring problem. To investigate how much our model, trained on SAT-50, can generalize to problems of different structures, we evaluate it on the flat graph coloring benchmark from SATLIB (Hoos & Stützle, 2000). All the problems in the benchmark are satisfiable.

Table 3 shows a decrease in GQSAT performance when generalizing to another problem distribution. We believe there are two potential reasons. First, different SAT problem distributions have different graph properties that are not captured during training on another distribution. Second, this might be related to our model selection process which does not favor generalization across problem structures.

Table 3 shows that graph coloring problems have more variables. We conducted an experiment investigating GQSAT's ability to scale to larger problems (more variables, more clauses). We trained GQSAT on flat75-180 with problems of 225 variables and 840 clauses. Graph Coloring benchmarks have only 100 problems each, so we do not split them into train/validation/test dataset using *flat-75-80* for training and *flat-100-239* to do model selection. We use the same hyperparameters as in all previous experiments changing only the gradient clipping parameter to 0.1. The results on Table 4 show that GQSAT can scale to bigger problems on the flat graph coloring benchmark.

Table 4: GQSAT Median Relative Iteration Reduction (MRIR) for the model trained on flat-75-80, evaluated on flat-100-239. flat-75-80 and flat-100-239 are separated, because the model was trained using the former and validated on the second. The results are for five training runs.

| dataset | GQSAT MRIR | | |
|---|---|---|---|
| | average | min | max |
| flat75-180 | 2.44 | 2.25 | 2.70 |
| flat-100-239 | 2.89 | 2.77 | 2.98 |
| flat-30-60 | 1.74 | 1.33 | 2.00 |
| flat-50-115 | 2.08 | 2.00 | 2.13 |
| flat-125-301 | 2.43 | 2.20 | 2.66 |
| flat-150-360 | 2.07 | 2.00 | 2.11 |
| flat-175-417 | 1.98 | 1.69 | 2.21 |
| flat-200-479 | 1.70 | 1.38 | 1.98 |

Apart from scaling to bigger graphs, we could test scaling for longer episodes. Table 1 shows exponential growth in the number of iterations it takes MiniSat to solve larger problems. Our preliminary experiments show that generalizing is easier than learning. Learning on SAT-100-430 requires more resources, does not generalize as well, and is generally less stable than training on SAT-50-218. This is most likely related to higher variance in the returns caused by longer episodes, challenges for temporal credit assignment, and difficulties with exploration, motivating further research. It also motivates curriculum learning as the next step of GQSAT development. Bapst et al. (2019) shows a positive effect of curriculum learning on RL with GNN.

## 4.3 DATA EFFICIENCY

We design our next experiment to understand how many different SAT problems GQSAT needs to learn from. We varied the SAT-50-218 train set from a single problem to 800 problems. Figure 3 demonstrates that GQSAT is extremely data efficient. Having more data helps in most cases but, even with a single problem, GQSAT generalizes across problem sizes and to unSAT instances. This should allow GQSAT to generalize to new benchmarks without access to many problems from it. We suppose that GQSAT's data efficiency is one of the benefits of using RL. The environment allows the agent to explore diverse state-action space regions making it possible to learn useful policies even from a single instance. In supervised learning, the data diversity is addressed at the training data generation step.

## 5 RELATED WORK

Using machine learning for the SAT problem is not a new idea (Haim & Walsh, 2009; Grozea & Popescu, 2014; Flint & Blaschko, 2012; Singh et al., 2009). Xu et al. (2008) propose a portfolio-based approach which yielded strong results in 2007 SAT competition. Liang et al. (2016) treat each SAT problem as a multi-armed bandit problem capturing variables' ability to generate learnt clauses.

Recently, SAT has attracted interest in the deep learning community. There are two main approaches: solving a problem end-to-end or learning heuristics while keeping the algorithm backbone the same. Selsam et al. (2018, NeuroSAT) take an end-to-end supervised learning approach demonstrating that GNN can generalize to SAT problems bigger than those used for training. NeuroSAT finds satisfying assignments for the SAT formulae and thus cannot generalize from SAT to unSAT problems. Moreover, the method is incomplete and might generate incorrect results, which is extremely important especially for unSAT problems. Selsam & Bjørner (2019) modify NeuroSAT and integrate it into popular SAT solvers to improve timing on SATCOMP-2018 benchmark. While the approach shows its potential to scale to large problems, it requires an extensive training set including over 150,000 data points. Amizadeh et al. (2018) propose an end-to-end GNN architecture to solve circuit-SAT problems. While their model never produces false positives, it cannot solve unSAT problems.

The following methods take the second approach learning a branching heuristic instead of learning an algorithm end-to-end. Jaszczur et al. (2019) take the supervised-learning approach using the same

graph representation as in Selsam et al. (2018). The authors show a positive effect of combining DPLL/CDCL solver with the learnt model. As in Selsam et al. (2018), their approach requires a diligent process of the test set crafting. Also, the authors do not compare their approach to the VSIDS heuristic, which is known to be crucial component of CDCL (Katebi et al., 2011).

Wang & Rompf (2018), whose environment we took as a starting point, show that DQN does not generalize for 20-91 3-SAT problems, whereas Alpha(GO) Zero does. Our results show that the issue is related to the state representation. They use CNNs, which are not invariant to variable renaming or permutations. Moreover, CNNs require a fixed input size which makes it infeasible when applying to problems with different number of variables or clauses.

The work of Lederman et al. (2018) is closest to ours. They train a REINFORCE (Williams, 1992) agent to replace the branching heuristic for Quantified Boolean Formulas using GNNs for function approximation. They note positive generalization properties across the problem size for problems from similar distributions. Besides the base RL algorithm and some minor differences, our approaches differ mainly in the state representation. They use 30 variables for the global state encoding and seven variables for vertex feature vectors. GQSAT does not require feature engineering to construct the state. We use only two bits to distinguish variables from clauses and encode literal polarities. Also, Lederman et al. (2018) use separate vertices for $x$ and $\neg x$ in the graph representation.

Vinyals et al. (2015) introduce a recurrent architecture for approximately solving complex geometric problems, such as the Traveling Salesman Problem (TSP), approaching it in a supervised way. Bello et al. (2016) consider combinatorial optimization problems with RL, showing results on TSP and the Knapsack Problem. Khalil et al. (2017) approach combinatorial optimization using GNNs and DQN, learning a heuristic that is later used greedily. It is slightly different from the approach we take since their heuristic is effectively the algorithm itself. We augment only a part of the algorithm – the branching heuristic. Paliwal et al. (2019) use GNN with imitation learning for theorem proving. Carbune et al. (2018) propose a general framework of injecting an RL agent into existing algorithms.

Cai et al. (2019) use RL to find a suboptimal solution that is further refined by another optimization algorithm, simulated annealing (Kirkpatrick et al., 1983, SA) in their case. The method is not limited with SA, and this modularity is valuable. However, there is one important drawback of the approach. The second optimization algorithm might benefit more from the first algorithm if they are interleaved. For instance, GQSAT can guide search before VSIDS overcomes its initialization bias.

Recently, GNN received a lot of attention in the RL community, enabling the study of RL agents in state/action spaces of dynamic size, which is crucial for generalization beyond the given task. Wang et al. (2018) and Sanchez-Gonzalez et al. (2018) consider GNN for the generalization of the control problem. Bapst et al. (2019) investigate graph-based representation for the construction task and notice high generalization capabilities of their agents. Jiang et al. (2018); Malysheva et al. (2018); Agarwal et al. (2019) study generalization of the behaviour in multi-agent systems, noting the GNN benefits due to their invariance to the number of agents in the team or other environmental entities.

## 6 CONCLUSION AND FUTURE WORK

In this paper, we introduced GQSAT, a branching heuristic of a SAT solver that causes more variable propagations per step solving the SAT problem in fewer iterations comparing to VSIDS. GQSAT uses a simple state representation and does not require elaborate reward shaping. We demonstrated its generalization abilities, showing more than 2-3X reduction in iterations for the problems up to 5X larger and 1.5-2X from SAT to unSAT. We showed how GQSAT improves VSIDS and showed that our method is data efficient. While our method generalizes across problem structures to a lesser extent, we showed that training on data from other distributions might lead to further performance improvements. Our findings lay the groundwork for future research that we outline below.

*Scaling GQSAT to larger problems.* Industrial-sized benchmarks have millions of variables. Our experiments training on SAT-100 and graph coloring show that increases in problem complexity makes our method less stable due to typical RL challenges: longer credit assignment spans, reward shaping, etc. Further research will focus on scaling GQSAT using latest stabilizing techniques (Hessel et al., 2018), more sophisticated exploration methods and curriculum learning.

*From reducing iterations to speeding up.* SAT heuristics are good because they are fast. It takes constant time to make a decision with VSIDS. GNN inference takes much longer. However, our experiments show that GQSAT can show an improvement using only the first $k$ steps. Reducing the network polling frequency and replacing the variable activities with GQSAT's output similarly to Selsam & Bjørner (2019) is another interesting avenue of the future research. An efficient C++ implementation of our method should also help.

*Interpretation of the results.* Newsham et al. (2014) show that the community structure of SAT problems is related to the problem complexity. We are interested in understanding how graph structure influences the performance of GQSAT and how we can exploit this knowledge to improve GQSAT.

Although we showed the powerful generalization properties of graph-based RL, we believe the problem is still far from solved and our work is just one stepping stone towards a new generation of solvers that can discover and exploit heuristics that are too difficult for a human to design.

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

## A    PROPAGATIONS PER STEP

## B    REPRODUCIBILITY

### B.1    MODEL ARCHITECTURE

We use Encoder-Process-Decode architecture from Battaglia et al. (2018). Encoder and decoder are independent graph networks, i.e. MLPs taking whole vertex or edge feature matrix as a batch without message passing. We call the middle part 'the core'. The output of the core is concatenated with the output of the encoder and gets fed to the core again. We describe all hyperparameters in Appendix B.3. We also plan to release the experimental code and the modified version of MiniSat to use as a gym environment.

### B.2    DATASET

We split SAT-50-218 into three subsets: 800 training problems, 100 validation and 100 test problems. For generalization experiments, we use 100 problems from all the other benchmarks.

For graph colouring experiments, we train our models using all problems from flat-75-180 dataset. We select a model, given performance on all 100 problems from flat-100-239. So, evaluation on this two datasets should not be used to judge the performance of the method and they are shown separately in Table 4. All the data from the second part of the table was not seen by the model during training (flat-30-60, flat-50-115, flat-125-301, flat-150-360, flat-175-417, flat-200-479).

### B.3    HYPERPARAMETERS

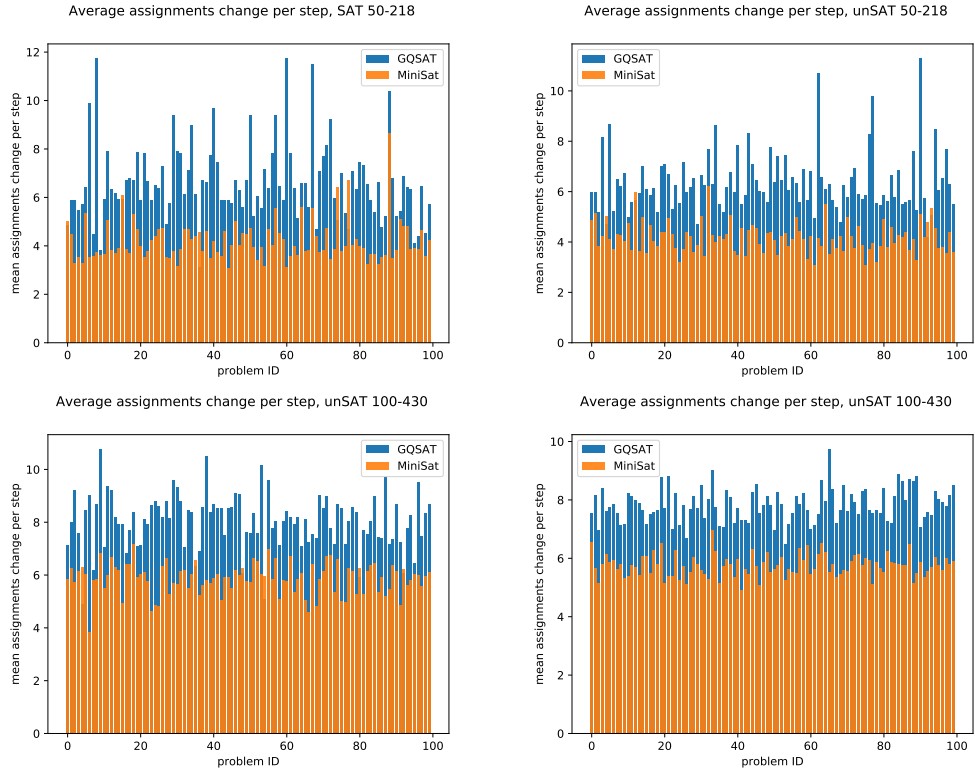

Figure 4: Average number of variable assignments change per step for (un)SAT-50-218 and (un)SAT-100-430.

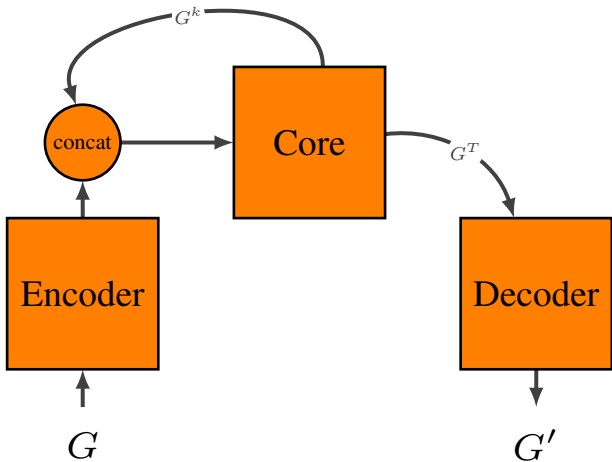

Figure 5: Encode-Process-Decode architecture. Encoder and Decoder are independent graph networks, i.e. MLPs taking whole vertex/edge data array as a batch. $k$ is the index of a message passing iteration. When concatenating for the first time, encoder output is concatenated with zeros.

| Hyperparameter | Value | Comment |
|---|---|---|
| *DQN* | | |
| – Batch updates | 50 000 | |
| – Learning rate | 0.00002 | |
| – Batch size | 64 | |
| – Memory replay size | 20 000 | |
| – Initial exploration $\epsilon$ | 1.0 | |
| – Final exploration $\epsilon$ | 0.01 | |
| – Exploration decay | 30 000 | Environment steps. |
| – Enitial exploration steps | 5000 | Environment steps, filling the buffer, no training. |
| – Discounting $\gamma$ | 0.99 | |
| – Update frequency | 4 | Every 4th environment step. |
| – Target update frequency | 10 | |
| – Max decisions allowed for training | 500 | Used a safety against being stuck at the episode. |
| – Max decisions allowed for testing | 500 | Varied among [0, 10, 50, 100, 300, 500, 1000] for the experiment on Figure 2. |
| – Step penalty size $p$ | -0.1 | |
| *Optimization* | | |
| – Optimizer | Adam | |
| – Adam betas | 0.9, 0.999 | Pytorch default. |
| – Adam eps | 1e-08 | Pytorch default. |
| – Gradient clipping | 1.0 | 0.1 for training on the graph coloring dataset. |
| – Gradient clipping norm | $L_2$ | |
| – Evaluation frequency | 1000 | |
| *Graph Network* | | |
| – Message passing iterations | 4 | |
| – Number of hidden layers for GN core | 1 | |
| – Number of units in GN core | 64 | |
| – Encoder output dimensions | 32 | For vertex, edge and global updater. |
| – Core output dimensions | 64,64,32 | For vertex, edge and global respectively. |
| – Decoder output dimensions | 32 | For vertex updater, since only Q values are used, no need for edge/global updater. |
| – Activation function | ReLU | For everything but the output transformation. |
| – Edge to vertex aggregator | sum | |
| – Variable to global aggregator | average | |
| – Edge to global aggregator | average | |
| – Normalisation | Layer Normalisation | After each GN updater |

