# OpenReview forum: "Improving SAT Solver Heuristics with Graph Networks and Reinforcement Learning"
_ICLR.cc/2020/Conference — Reject_

### Official Review · AnonReviewer3 · 2019-10-20
**Official Blind Review #3**

**Rating:** 8

**Review:**

This paper investigates the problem of learning new branching heuristics in SAT solvers. The idea is very simple: take MiniSat, remove the usual VSIDS heuristic, and replace it with a variable selection policy that has been trained from a deep reinforcement learning algorithm. The architecture advocated in the present study is based on GNNs coupled with usual DQN techniques. The resulting GQSAT heuristic is endowed with attractive properties: on random SAT instances, it outperforms VSIDS and generalizes relatively well to other SAT distributions.

Overall, this is a very interesting paper. It is well-written, well-motivated, and well-positioned with respect to related work. The conceptual idea is simple and elegant, the choice of the graph-based DQN architecture is relevant, the experimental protocol is well-detailed, and the experimental results look promising. To sum up, I have no major reasons for not accepting this paper.

Some potential improvements:

(a) Obviously, the decision model for GQSAT is an “episodic” MDP. It would be relevant to emphasize this aspect by presenting episodic MDPs (instead of standard ones) in Sec 2.2.
(b) States representations only encode SAT formulae (using Q-labeled incidence graphs). Although this is a conceptually simple idea, GQSAT could exploit additional information provided by MiniSAT (e.g. number of propagations, number of clauses which have been learned, etc.). I am wondering whether such “solver features” in state representations could improve the GQSAT heuristic, and could help in generalizing from a class of SAT problems to another one.
(c) The notion of "terminal state" is a bit ambiguous. Since the number of actions per episode is capped, a terminal state can be a leaf of the MiniSat search tree (where a satisfying assignment was found, or a dead-end was reached), or an internal node of the tree (when the maximum number of actions per episode was reached).


**Experience Assessment:**

I have read many papers in this area.

**Review Assessment: Checking Correctness Of Derivations And Theory:**

I assessed the sensibility of the derivations and theory.

**Review Assessment: Checking Correctness Of Experiments:**

I assessed the sensibility of the experiments.

**Review Assessment: Thoroughness In Paper Reading:**

I read the paper at least twice and used my best judgement in assessing the paper.

---

> ### Author Response · Authors · 2019-11-11
> **Author reply**
>
> Thanks a lot for your feedback!
>
> We will update the paper to incorporate your suggestion on episodic MDPs and will clarify the notion of the terminal state.
>
> Investigating the state representation further is a very important direction of future work, we agree, that there is a lot of interesting unanswered questions here.  In general, adding more relevant information should help, however, it might also impoverish generalisation capabilities of our method or affect it sample efficiency. The reason for this is that when we use less information in the observation, effectively we have more data to learn and generalise from. Using local observation instead of full global state has been known to positively affect an RL agent performance.

---

### Official Review · AnonReviewer1 · 2019-10-23
**Official Blind Review #1**

**Rating:** 3

**Review:**

=== Summary ===
The authors apply Q-learning with GNN function approximation as a branching heuristic in Conflict Driven Clause Learning (CDCL) SAT solvers. The GQSAT agent assigns unassigned variables to True or False in a CDCL SAT environment and receives a negative reward for each non-terminal state, thus encouraging the agent to finish an episode as quickly as possible.
The agent is trained with Q-learning and GNN as the function approximator: the input is presented as a bipartite graph with variables and clauses corresponding to a SAT instance in CNF form and the GNN predicts Q-values for assigning each unassigned variable to True or False.
Then authors compare their method to the popular VSIDS branching heuristic (which counts the number of conflicts a literal or variable has been involved in) and report the median relative iteration reduction over all problems (~2-3x for satisfiable instances, a bit less for unsatisfiable instances).
GQSAT generalizes well to larger satisfiable instances than those seen during training and somewhat to unsatisfiable instances (unSAT). It doesn't seem to provide improvements compared to VSIDS when when generalizing to different problem structures (as shown by the graph coloring generalization experiments).

=== Recommendation ===

This paper is well-motivated (improving speed of complete SAT solvers) and presents a good overview of related work.
The paper is well written overall, although I found section 3 hard to read and I think the writing can be a bit improved by writing explicitly the algorithm or at least the MDP (see misc comments)
While the reduction in number of branching decisions is satisfactory, this does not necessarily translates to actual speed-ups (since the VSIDS branching heuristic is faster in practice) and there are some concerns about applying GQSAT to larger instances.

I slightly lean for rejecting this paper.

=== Misc ===
- Can the authors hypothesize on why GQSAT does not benefit from seeing more instances of SAT during training?
- What is the motivation for using Median Relative Iteration Reduction as a metric? How about using the ratio of the means over all problems instead?
- Beginning of section 3 is quite confusing: "We represent the set of all possible SAT problems as an MDP. The state of such an MDP consists of unassigned variables and unsatisfied clauses.". It would help to write properly the MDP corresponding to the CDCL SAT solver.
- 'NeuroSAT cannot generalize from SAT to unSAT': the NeuroSAT paper showed that they could learn to predict satisfiability as well.

**Experience Assessment:**

I have read many papers in this area.

**Review Assessment: Checking Correctness Of Derivations And Theory:**

I assessed the sensibility of the derivations and theory.

**Review Assessment: Checking Correctness Of Experiments:**

I assessed the sensibility of the experiments.

**Review Assessment: Thoroughness In Paper Reading:**

I read the paper at least twice and used my best judgement in assessing the paper.

---

> ### Author Response · Authors · 2019-11-11
> **Author Reply**
>
> Thanks a lot for your feedback!
>
> Your main concerns arise from the viewpoint of the practical application of GQSAT for the industrial-sized problems/benchmarks. Since ICLR is a machine learning venue, our goal is not to build a SAT competition contender but rather ‘Can we apply RL to such a challenging problem as a branching heuristic of a SAT solver and what are the RL challenges we need to overcome?’.  The metric we use is good enough to understand if RL does something sensible and study our method's generalisation and data efficiency properties.
>
> There are a lot of RL challenges we addressed: dealing with a changing state-action space (graph networks), dealing with a novel benchmark for RL (a completely new set of hyperparameters, environment optimization, state space encoding), learning from a simplistic state representation and sparse reward. We also believe, that studying generalization properties of an RL agent across several dimensions is an important research result on its own. In our work, we were largely inspired by NeuroSAT, which asked similar scientific questions and studied them on small problems as well. NeuroSAT, accepted at ICLR last year, did not deal with the scaling issues and did not have an obvious industrial application since it was not a complete algorithm (it might give a wrong answer). Later, the authors amended the method (named NeuroCore) and published it at a SAT conference, focusing on the application side.

---

> > ### Comment · AnonReviewer1 · 2019-11-14
> > **Reviewer reply**
> >
> > I respectfully disagree with the author reply.
> >
> > I do not expect from the authors a SAT competition contender or a viable industrial alternative but I believe that considering how the approach scales is central in evaluating the method.
> >
> > "There are a lot of RL challenges we addressed": I do not see this as a strength of the paper. If anything, this could indicate that RL (or the method proposed here) is not the right approach for this problem, since the experimental results aren't that significant, especially considering actual speed and scaling issues.

---

> > > ### Author Response · Authors · 2019-11-15
> > > **Author Reply**
> > >
> > > We believe that our experimental results are useful for the ICLR community and do not agree that the experimental results "aren't that significant". GQSAT data efficiency and zero-shot generalisation to the problems 5x larger in state-action space size and even more (if we consider the horizon length as the measure of complexity) is an important result for an RL algorithm.
> > >
> > > We also updated the paper addressing your 'misc' section comments.

---

### Official Review · AnonReviewer2 · 2019-10-24
**Official Blind Review #2**

**Rating:** 3

**Review:**

The paper proposes learning a branching heuristic to be used inside the SAT solver MiniSat using reinforcement learning. The state is represented as a graph representation of the Boolean formula as in previous works, and the policy is parameterized as a graph neural network. At each step of an episode the policy selects a variable to branch on and assigns a value to it. The episode terminates once the solver finds a satisfying assignment or proves unsatisfiability. The reward function encourages the policy to reach terminal state in as few steps as possible. The policy is trained using DQN. Results on randomly generated SAT instances show that the learned policy is able to solve problems with fewer steps than VSIDS, the branching heuristic commonly used by state-of-the-art solvers.

Pros:
- The paper is nicely written and easy to read.
- The general idea of using RL to learn distribution-specific branching heuristics is a very interesting research problem, and SAT is a difficult test case for it.
- The experiments provide interesting insights, especially Figure 2 and the graph coloring results in Table 3.

Cons:
- Showing improvements in the number of steps compared to VSIDS is not interesting because VSIDS as implemented in MiniSat and state-of-the-art solvers like Glucose has been tuned to minimize running time rather than number of steps.  A better comparison would be to compare against a branching heuristic that is designed to be step-efficient -- e.g., the branching heuristic GGB proposed in Chapter 3 of Liang’s PhD thesis (available here: https://drive.google.com/file/d/1RzJtmdbjFeT2N84WDWQkBfQoGPF-qRoT/view?usp=sharing). GGB is more expensive than VSIDS, but if the time needed for branching is removed, a solver using GGB is faster than one using VSIDS (see figure 3.1 in the thesis).

- A better discussion of how to scale up the proposed approach to instances with millions of variables is needed. Although most ML papers on SAT deal with at most hundreds of variables, such small instances are trivial for the state-of-the-art solvers. The real challenge is to scale up to the instance sizes that are considered in SAT competitions. For example, Selsam and Bjorner 2019 https://arxiv.org/pdf/1903.04671.pdf tackle such instances in their NeuroCore work. (While the paper references that work, it doesn’t compare to it.) There is no attempt to address the scalability issue in this work. Apart from considering the challenges of successfully learning on large instances (briefly discussed in the conclusion), there should be an analysis on the inference cost of the graph neural network (which scales linearly with the number of variables and clauses) to make a single branching decision vs. that of VSIDS, and what that implies on how much reduction in the number of steps a learned branching policy would need to achieve before it can provide time savings over VSIDS. It may turn out that the reductions required of a learned branching policy are implausibly large. Without a better understanding of these challenges, it is not clear that learning can help much to improve the state-of-the-art SAT solvers.

**Experience Assessment:**

I have read many papers in this area.

**Review Assessment: Checking Correctness Of Derivations And Theory:**

I assessed the sensibility of the derivations and theory.

**Review Assessment: Checking Correctness Of Experiments:**

I assessed the sensibility of the experiments.

**Review Assessment: Thoroughness In Paper Reading:**

I read the paper at least twice and used my best judgement in assessing the paper.

---

> ### Author Response · Authors · 2019-11-11
> **Author Reply**
>
> Thanks a lot for your feedback!
>
> Both of your main concerns are related to the practical application of our approach for the industrial-sized problems. Since ICLR is a machine learning venue, for us, the main question was mostly about whether we can stretch existing RL techniques to learn a branching heuristic of a SAT solver and what potential difficulties we might face. As you pointed out, SAT is a very difficult test case for RL here.
>
> Using #of steps as a metric would not be 'interesting' if we claimed GQSAT is ready for the industrial application. However, as we mention above, we are interested in RL behaviour using SAT branching heuristic as a testbed. In this scenario, the metric is good enough to evaluate the relative performance of the method, study its generalization and data efficiency properties, and identifying potential drawbacks and directions for future work.
>
> We agree that scaling concerns would be an integral part of the paper claiming state-of-the-art performance for the purposes of the SAT-solving community. The mentioned NeuroCore work which was published at an applied SAT conference is based on NeuroSAT, published last year at ICLR. NeuroSAT did not have scalability analysis studying the properties of their approach in a non-realistic setting. We were inspired by NeuroSAT (accepted at ICLR last year). We believe that our experiments on generalization and data efficiency is an important result on its own, and, as you mentioned, the insights we provide are interesting and sometimes surprising.
>
> We believe that the scaling properties of our approach will not be very different from NeuroCore if we speak about the inference. The inference phase for a model trained with RL and supervised learning is the same. Both approaches use similar networks (shallow message-passing networks) and should scale similarly. As we can see in the NeuroCore example, the method saves some wall-clock time periodically querying the network and replacing the variable activities using its predictions. We do not see any significant obstacles to a similar usage of GQSAT. However, this will require some engineering work which was out of the scope of the current research.

---

### Public Comment · ~Mislav_Balunovic1 · 2019-10-13
**Related paper and a question**

Hi, it is great to see more work in this research area. We have also recently published a paper [1] in NeurIPS 2018 where we proposed a way to learn heuristics which improve the performance of an SMT solver (SMT is generalization of SAT considered in your work). The problem we faced is that SMT formulas generally have more complex structure and larger size than SAT formulas which is why we used word embeddings to build feature representation of these formulas. Do you think it would be possible to use your approach to instead learn feature representation of the SMT formulas using graph neural networks?

[1] Mislav Balunovic, Pavol Bielik, and Martin Vechev. "Learning to solve SMT formulas." Advances in Neural Information Processing Systems. 2018.

---

> ### Author Response · Authors · 2019-10-15
> **Graph Networks and RL for SMT**
>
> Graph Neural Networks is a relatively new area. We believe that they have a lot of potential and might be useful in learning the local structure of your problem to extract features useful for the solver. There is some work on constraint satisfaction solvers using graph networks [1] and applying Q-learning [2], which might be of interest to you.
>
> [1] Amizadeh, Saeed, Sergiy Matusevych, and Markus Weimer. "PDP: A General Neural Framework for Learning Constraint Satisfaction Solvers." arXiv preprint arXiv:1903.01969 (2019).
> [2] Xu, Yuehua, David Stern, and Horst Samulowitz. "Learning adaptation to solve constraint satisfaction problems." Proceedings of Learning and Intelligent Optimization (LION) (2009).

---

### Decision · Program_Chairs · 2019-12-19

**Decision:**

Reject

**Comment:**

SAT is NP-complete (Karp, 1972) due its intractable exhaustive search. As such, heuristics are commonly used to reduce the search space. While usually these heuristics rely on some in-domain expert knowledge, the authors propose a generic method that uses RL to learn a branching heuristic. The policy is parametrized by a GNN, and at each step selects a variable to expand and the process repeats until either a satisfying assignment has been found or the problem has been proved unsatisfiable. The main result of this is that the proposed heuristic results in fewer steps than VSIDS,  a commonly used heuristic.

All reviewers agreed that this is an interesting and well-presented submission. However, both R1 and R2 (rightly according to my judgment) point that at the moment the paper seems to be conducting an evaluation that is not entirely fair. Specifically, VSIDS has been implemented within a framework optimized for running time rather than number of iterations, whereas the proposed heuristic is doing the opposite. Moreover, the proposed heuristic is not stressed-test against larger datasets. So, the authors take a heuristic/framework that has been optimized to operate specifically well on large datasets (where running time is what ultimately makes the difference) scale it down to a smaller dataset and evaluate it on a metric that the proposed algorithm is optimized for. At the same time, they do not consider evaluation in larger datasets and defer all concerns about scalability to the one of industrial use vs answering ML questions related to whether or not it is possible to  “stretch existing RL techniques to learn a branching heuristic”. This is a valid point and not all techniques need to be super scalable from iteration day 0, but this being ML, we need to make sure that our evaluation criteria are fair and that we are comparing apples to apples in testing hypotheses. As such, I do not feel comfortable suggesting acceptance of this submission, but I do sincerely hope the authors will take the reviewers' feedback and improve the evaluation protocols of their manuscript, resulting in a stronger future submission.